# Effects of Node Centrality Measures for Road Type Classification using Graph Neural Networks

Anonymous Full Paper
Submission 43

## Abstract

This study explores the impact of feature selection, particularly node centrality measures, on road type classification within a road network graph using Graph Neural Networks (GNNs) and traditional machine learning models. By training six models on three distinct feature sets—primary road characteristics (S1), centrality measures (S2), and a combined feature set (S3)—we analyze how different feature representations affect model accuracy in distinguishing road types. The GraphSAGE model using S1 achieved the highest test accuracy (0.89), indicating that primary road characteristics are highly effective for classification, whereas the Random Forest model performed worst on the same set, achieving only 0.17 accuracy. Visualized embeddings from S1 models reveal effective clustering by road type for models like GraphSAGE, particularly for residential and tertiary roads, underscoring the model's capability to capture nuanced structural relationships. These findings indicate that feature selection, especially the inclusion of relevant node centrality measures, plays a crucial role in enhancing classification, though further improvement may require hybrid models or additional contextual data sources to address limitations in differentiating road types with overlapping attributes.

## 1 Introduction

Graph Neural Networks (GNNs) offer a robust framework for developing deep neural networks tailored to graph data. A key feature of GNNs is their use of neural message passing, where vector messages are communicated between nodes and processed through neural networks [1, 2]. The motivation for employing GNNs arises from the shortcomings of traditional neural networks, which typically function best with Euclidean data structured in regular grids, such as images (2D pixel grids) or sequences (1D time-ordered arrays). In such cases, data relationships are often implicit or follow a predetermined pattern, as seen with adjacent pixels in images or sequential time steps in data. However, many real-world scenarios present non-Euclidean data, where entities exhibit complex and irregular interconnections that cannot be easily organized into grids or arrays. A good example is a road network, where intersections serve as nodes and the roads as edges. Unlike the structured nature of images or sequences, road networks feature irregular connections—some intersections may link to multiple roads, while others could connect to just one or two [3]. Additionally, the distances between intersections may vary, and the relationships among roads and intersections often carry crucial information, such as traffic dynamics, optimal routes, or road conditions. The existing literature categorizes the application of GNNs in graphs into three main tasks: graph-level, edge-level, and node-level tasks [4]. Researchers have effectively harnessed GNNs for various applications within road networks, including tasks such as road surface extraction [5–7], traffic prediction [8–10], and road crack detection [11]. In particular, road type classification has benefited from edge-based [12] and node-based approaches [13], illustrating the versatile capabilities of GNNs in addressing real-world challenges. GNN architectures can be categorized into three main types: convolutional mechanisms, attention mechanisms, and autoencoder mechanisms [4]. The convolutional mechanism, typically referred to as GCN, employs convolutional or pooling operations on graph structures. This method effectively extracts richer representations for each node, which can be applied to node classification tasks. However, a limitation of this approach is its inherently transductive nature; it requires the presence of all nodes during training, making it difficult to generalize to unseen nodes. This challenge led to the development of GraphSAGE [14], which introduces an inductive model capable of accommodating new nodes, thus enhancing its applicability. In contrast, attention mechanisms differentiate themselves from GCN-based models by assigning variable contributions from different neighboring nodes to the target node. This allows the model to concentrate on the most relevant information, improving the overall effectiveness of the classification process. The utility of the autoencoder mechanism lies in its ability to facilitate unsupervised learning, enabling the creation of low-dimensional embeddings from large sets of unlabeled training data [1]. Despite the competitive performance of GNNs in classifying graph-structured data, it is crucial to emphasize feature selection to maintain high performance [15]. In real-world applications, features of neighboring nodes across different hops may not always correlate with

the target node's features, leading to potential noise in the model's aggregation process. Additionally, addressing the challenge of imbalanced datasets is essential, as such imbalances can significantly impact model performance [16]. By focusing on these aspects, we can further enhance the effectiveness and robustness of GNN implementations in various applications. Through the presented *Graph Neural Networks* (GNN) arhictectures, this study aims to observe the performance associated when road network graphs are used as input for node classification task, especially road network may have a property of homophily and heterophily [17]. To be specific, the study aims to achieve the following objectives:

1. Train four GNN models using three sets of features: (1) primary road network characteristics: number of intersections connecting the road, speed limit, length, number of lanes, and oneway attribute denoting if the road accepts one or bidirectional traffic, (2) node centrality measures, and (3) the primary road network features and node centrality measures.

2. Train four baseline machine learning models using the same feature set in the previous item.

3. Compare the performance of these models across these three feature sets with other baseline machine learning models.

This work offers a novel contribution by exploring a unique set of objectives that, to the best of the researchers' knowledge, has not been comprehensively studied before. Specifically, it focuses on feature-engineered node attributes within road networks, including various centrality measures. Additionally, there has been limited investigation into using road network information for node classification of road types with the proposed feature sets. While mathematical graphs have a universal definition, the specific context and characteristics of road network data present challenges when adapting deep learning architectures that have been successful in other domains. This highlights the importance of analyzing road network data in this particular context. Furthermore, the research by [18] emphasizes the potential benefits of incorporating additional features, such as road lanes, to enhance predictive accuracy.

## 2 Methods

### 2.1 The Road Network Data

This study focuses on extracting road network information from selected regions within the National Capital Region of the Philippines using OpenStreetMap (OSM). It highlights an observed class imbalance regarding highway types. In contrast to the approach taken by [13], which involved recategorizing labels into different classes, we address this imbalance by selectively choosing a subset of highway categories: `residential`, `tertiary`, `secondary`, `primary`, and `unclassified`. To enhance the quality of our analysis, we preprocess the data to ensure that the resulting graph is undirected, connected, and simple, thereby eliminating any multiple edges. Following this, we employ Geographic Information Systems (GIS) to streamline the graph partitioning process, which is essential for organizing the training, testing, and validation sets, as illustrated in Figure 1. This method ensures that each edge set remains mutually exclusive.

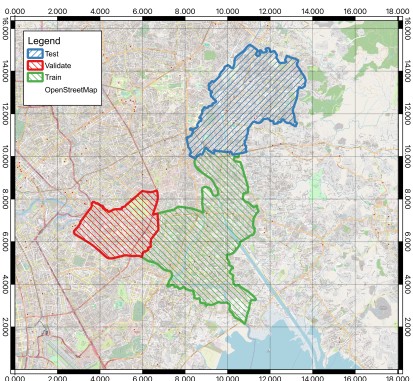

**Figure 1.** The Road Network Data.

| Data | Streets | Nodes |
|---|---|---|
| Train | 6009 | 4551 |
| Test | 4027 | 2868 |
| Validate | 1707 | 1253 |

**Table 1.** Street and Intersection Count of Train, Test, Validate Sets.

This study explores three distinct input feature vectors to enhance our understanding of road dynamics. The first vector focuses on primary road characteristics, incorporating elements such as `maxspeed`, `length`, `oneway`, `intersection_count`, and `number of lanes`. The second vector delves into node centrality measures, which include `degree centrality`, `betweenness centrality`, and `closeness centrality`. Additionally, we examine the effects of combining both feature sets to provide a more comprehensive analysis. All numeric features (except oneway) are standardized which transforms the data:

$$x_i' = \frac{x_i - \mu}{\sigma} \qquad (1)$$

where $\mu$ is the mean, $\sigma$ is the standard deviation of the features in the training set. In order to process the `networkx` graph it needs to be first converted into the format used by `PyTorch Geometric`, which

requires the node feature matrix, edgelist, and labels of each node.

## 2.2 The GNN Pipeline

This study uses four GNN-based models trained and evaluate separately: *Graph Convolutional Networks* (GCN) [19], ChebNet [20], *Graph Attention Networks* (GAT), and GraphSAGE and four baseline models: *Random Forest* (RF), *Support Vector Classifier* (SVC), *Naive Bayes Classifier* (NB), and *Recurrent Neural Networks* (RNN). The GNN models follows the architecture depicted in Figure 2.

For the first model, once the graph features are prepared, they are passed into a GCN layer where for each node $v$, the model aggregates the features of its neighbors and transforms them using learnable weights. The layer output is computed as:

$$\mathbf{H}^{(l+1)} = \sigma\left(\tilde{\mathbf{A}}\mathbf{H}^{(l)}\mathbf{W}^{(l)}\right) \tag{2}$$

$\mathbf{H}^{(l)}$ is the feature matrix of the nodes at layer $l$, $\tilde{\mathbf{A}}$ is the normalized adjacency matrix, which is computed as:

$$\tilde{\mathbf{A}} = \mathbf{D}^{-\frac{1}{2}}(\mathbf{A} + \mathbf{I})\mathbf{D}^{-\frac{1}{2}} \tag{3}$$

such that $\mathbf{I}$ is the identity matrix and $\mathbf{D}$ is the degree matrix. $\mathbf{W}^{(l)}$ is the learnable weight matrix for layer $l$ and $\sigma$ is the activation function. Note that the first layer accepts the data features as input given as:

$$\mathbf{H}^{(0)} = \mathbf{X} \tag{4}$$

The input layer receives the feature matrix and edge index, which represent the dual graph nodes and their connections, respectively. The first hidden layers applies the first graph convolution operation, which makes use of the `eLU` activation and dropout as displayed in equation 5:

$$\mathbf{H}^{(1)} = \text{Dropout}\left(\text{eLU}\left(\tilde{\mathbf{A}}\mathbf{X}\mathbf{W}^{(0)}\right)\right) \tag{5}$$

to produce the hidden representation at layer 1, for which is taken by layer 2 where a second graph convolution is applied to produce the logit matrix $\mathbf{Z}^{|\mathcal{V}| \times C}$, where $|\mathcal{V}|$ is the number of nodes and $C$ is the number of classes, computed using equation 6:

$$\mathbf{Z} = \tilde{\mathbf{A}}\mathbf{H}^{(1)}\mathbf{W}^{(1)} \tag{6}$$

The output layer then applies the `log softmax` function to the logits to produce the final class probabilities. When dealing with large or small logits, computing the softmax function directly can lead to numerical instability due to the exponentiation of these values. This can result in overflow (very large numbers) or underflow (very small numbers approaching zero), which can cause NaN (not a number) results, which makes log softmax function more advantageous in some cases. The model ChebNet

[20], a generalization of the GCN framework, is also used that applies Chebyshev convolution layers to the node features, mathematically executed using the equation:

$$\mathbf{x}^{(l+1)} = \sigma\left(\sum_{k=0}^{K} \theta_k \mathbf{T}_k(\tilde{\mathbf{L}})\mathbf{x}^{(l)}\right) \tag{7}$$

where: $\mathbf{x}^{(l)}$ is the node feature matrix at layer $l$, $\mathbf{T}_k(\tilde{\mathbf{L}})$ are Chebyshev polynomials applied to the normalized Laplacian $\tilde{\mathbf{L}}$, $\theta_k$ are learnable parameters, $\sigma$ is a non-linear activation function (ReLU in this case), and $K$ is the polynomial order. The network used in this study is composed of two layers of Chebyshev convolution, with dropout applied after the first layer to prevent overfitting. The ability of ChebNet to compute Chebyshev polynomials of the normalized graph Laplacian $\tilde{\mathbf{L}}$ in linear time relative to the polynomial order $K$ represents a significant advantage over traditional spectral graph convolutional networks (GCNs), which rely on computationally expensive spectral decompositions. While GCNs face a complexity of $O\left(n^3\right)$ due to the need to evaluate eigenvalues and eigenvectors of the graph Laplacian, ChebNet uses Chebyshev polynomial approximations to express graph convolutions as polynomial evaluations without requiring a full spectral decomposition. This is facilitated by the recursive definition of Chebyshev polynomials, given as: Base Cases:

$$\mathbf{T}_0(x) = 1 \quad \text{(constant polynomial)}$$
$$\mathbf{T}_1(x) = x \quad \text{(linear polynomial)}$$

Recursive Relation: For $k \geq 1$ :

$$\mathbf{T}_{k+1}(x) = 2x\mathbf{T}_k(x) - \mathbf{T}_{k-1}(x)$$

which allows for efficient computation by building on previously calculated values, resulting in only $O(K)$ computations for $K$ polynomials. The main operations involve multiplying the graph Laplacian by these polynomials, yielding an overall complexity of $O(K \cdot m)$, where $m$ is the number of edges in the graph. Consequently, ChebNet effectively harnesses the graph structure to aggregate information across both local and distant node relationships while significantly reducing computational overhead. Two other GNN models are trained, the GAT and GraphSAGE. GAT introduces the concept of attention to graph neural networks, allowing the model to assign different weights to neighbors based on their relevance to a given node. This is achieved by learning attention coefficients, which determine the importance of neighboring node features when aggregating information. After computing these coefficients, GAT aggregates information from neighboring nodes by taking a weighted sum of their features, allowing the network to focus on more relevant neighbors. GraphSAGE, in contrast, follows a neighborhood sampling

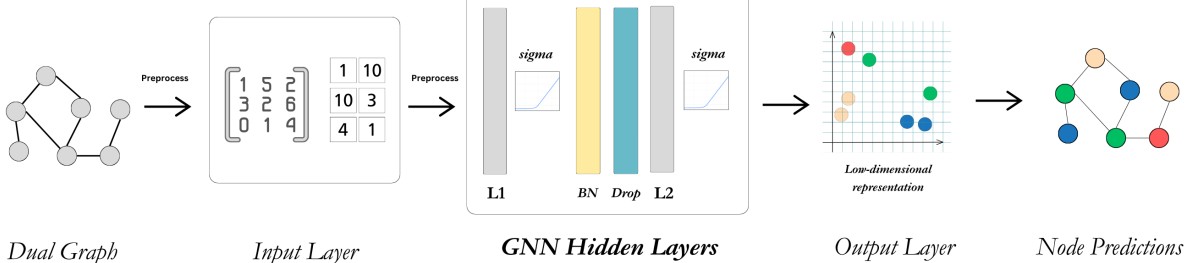

**Figure 2.** The GNN Pipeline

and aggregation approach, where each node aggregates information from a sampled set of its neighbors, making it particularly useful for inductive settings where new nodes are encountered during testing. GraphSAGE aggregates neighborhood information using functions such as mean, LSTM-based, or pooling aggregators, represented as:

$$\mathbf{h}_i^{(l+1)} = \sigma \left( \mathbf{W}^{(l)} \cdot \text{AGG} \left( \mathbf{h}_i^{(l)} \cup \mathbf{h}_j^{(l)}, \forall j \in \mathcal{N}(i) \right) \right) \tag{8}$$

where $\mathbf{h}_i^{(l+1)}$ represents the node embeddings at layer $l+1$, $\mathbf{W}^{(l)}$ is a layer-specific learnable weight matrix, and AGG is the chosen aggregation function, in this case the mean function. The performance metrics include accuracy, precision, recall, F1-score and will be compared across different feature sets.

## 3    Results and Discussion

This section discusses the results obtained from training six machine learning models for the node classification task using a road network as the input graph. The analysis reveals significant differences in performance across various models and feature sets, emphasizing the importance of feature selection in machine learning applications. The GraphSAGE model utilizing feature set S1, which focuses on primary road characteristics, achieved an impressive test accuracy of 0.89, as shown in Table 2. In contrast, the Random Forest (RF) model recorded the worst accuracy of 0.17 with the same feature set. It is important to highlight the effects of changing the feature set for each model. For instance, the best-performing model for S1, D4, exhibited a drastic drop in accuracy when tested with node centrality features only (S2) and combination (S3), achieving only 0.48 and 0.59 accuracy, respectively. This suggests that centrality measures may not adequately capture the rich contextual information and combining it with road features may not be sufficient. Notably, there is an increase in performance for RF, SVC, RNN when S2 and S3 are used in training, although their accuracy are still inferior compared to the best values found in each feature set. While

the Naive Bayes (NB) model is not based on Graph Neural Networks (GNNs), its performance proves to be competitive with that of the Graph Attention Network (GAT). The node embeddings obtained from S1 models are reduced into 2 components using t-SNE and are visualized in Figure 3. The figure shows the comparison of various models used for road type classification, with each road segment colored according to the ground-truth classification of the embedding: primary, residential, secondary, tertiary, and unclassified.

In general, GNN-based models exhibit dense and cohesive clustering patterns across various road types, with particularly close associations among residential roads. However, the GCN and GAT models face challenges in distinguishing tertiary roads from other highway types. Although ChebNet achieves lower intra-cluster distances than GCN and GAT, GraphSAGE produces better separation across residential, tertiary, primary, and unclassified road types. Despite these improvements, all GNN models struggle to separate tertiary roads from residential roads, primarily because these road types share similar physical characteristics. The distinction arises more from functional aspects and contextual elements, such as infrastructure and nearby buildings; for instance, while both road types share similar traits, residential roads primarily serve neighborhoods and subdivisions. GraphSAGE also encounters difficulty distinguishing secondary highways, often overlapping with primary and unclassified roads. This observation aligns with the role of secondary roads, which, while not as critical as primary highways, are integral to national and local route networks. In urban areas, secondary roads often serve as major arteries with characteristics similar to primary roads, such as lane count and intersection frequency, but are distinguished by surrounding infrastructure and the topological layout of the city. Unlike GNN-based models, traditional machine learning models show poorer separation between ground-truth road labels, often classifying road types into more than five classes due to more dispersed embeddings. This indicates that traditional models struggle to capture the graph's structural relationships effectively.

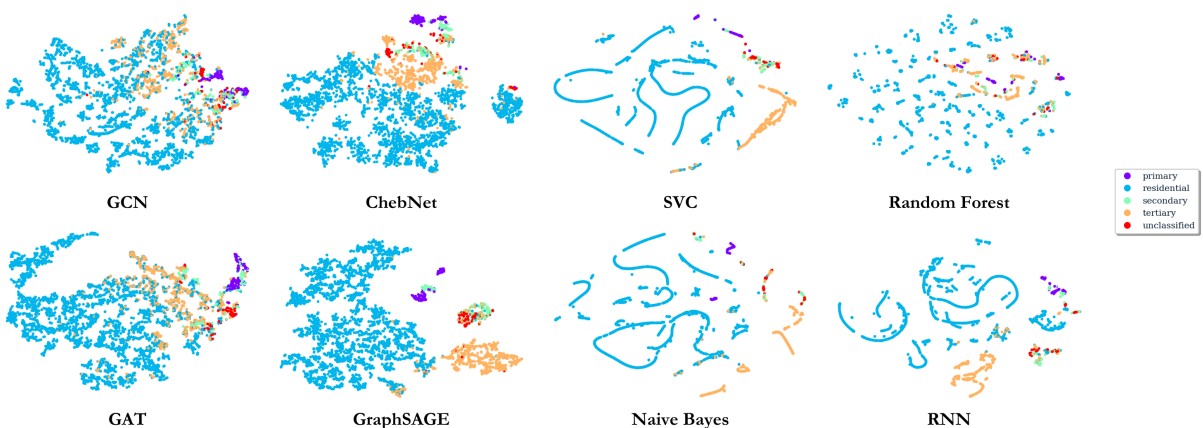

**Figure 3.** S1 Feature Set: Visualization of the node embeddings learned by the different models.

**Table 2.** Accuracy Results for S1, S2, and S3 Across All Models

| Model | S1 | S2 | S3 |
|---|---|---|---|
| **D1 (GCN)** | 0.70 | 0.76 | 0.72 |
| **D2 (ChebNet)** | 0.80 | 0.75 | 0.74 |
| **D3 (GAT)** | 0.79 | 0.62 | 0.74 |
| **D4 (GraphSAGE)** | **0.89** | 0.48 | 0.59 |
| **B1 (RF)** | 0.17 | 0.43 | 0.44 |
| **B2 (SVC)** | 0.21 | 0.76 | 0.74 |
| **B3 (NB)** | 0.77 | 0.59 | 0.61 |
| **B4 (RNN)** | 0.12 | 0.52 | 0.54 |

Another analysis is conducted by focusing on the S1, the superior feature set, through the F1, Precision, Recall, Macro, and Weighted averages across all models as shown in 4 and 5. For GCN, the macro avg and weighted avg show an imbalance in the prediction performance across classes, with relatively high performance for the residential class (precision = 0.90, recall = 0.79), but poor for the other categories like secondary (precision = 0.29) and unclassified (precision = 0.00). The ChebNet has improved accuracy and better performance in classes like tertiary (precision = 0.61, recall = 0.69), balanced performance between classes compared to D1, as seen in the macro average, but still struggling with unclassified. The precision remains low for unclassified label in D3. In D4, the model has significantly improved predictions for tertiary (F1-score = 0.83) and strongest performance on the residential class (F1-score = 0.97). In general, it is seen that there is a persistent struggle with the unclassified class in all GNN models. In terms of baseline models, B1 and B2 fails to generalize well to other categories: B1 seems to overfit the primary class (precision = 1.00) and B2 performs best on secondary roads (recall = 1.00), though at the cost of low precision in many classes. Like the GNN models, the NB struggles with unclassified roads.

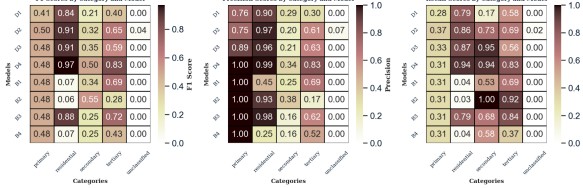

**Figure 4.** S1 Feature Set: Precision, Recall, and F1 Scores for different classes across all models.

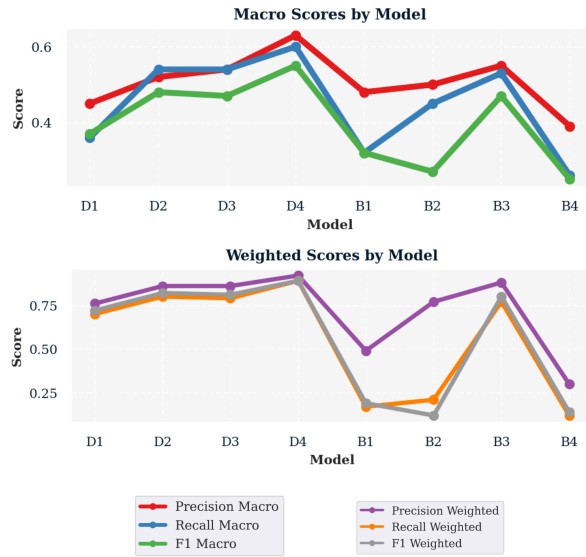

**Figure 5.** S1 Feature Set: Macro and Weighted Average of Precision, Recall, F1 scores across all models.

Finally, the behavior of the two superior models on S1, the ChebNet and GraphSAGE are analyzed through modifying the essential mode parameters and its effect with respect to running time and accuracy. According to Figure 6, increasing the number of hidden dimensions in GraphSAGE generally leads to better accuracy, but with some fluctuations. This suggests that higher hidden dimensions may allow the model to capture more complex relationships be-

tween road properties (features) and highway types but could also mean that beyond a certain complexity, the model might start overfitting or failing to generalize well to certain node types and the increase in hidden dimensions does not directly lead to higher accuracy, given that the increase is only of minute places. As hidden dimensions increase, training time is relatively stable within seven to nine seconds of running time. On the other hand, the same figure shows the result how the polynomial degree of the Chebyshev polynomial affects the ChebNet accuracy and running time. The Chebyshev polynomial degree determines how far the influence of each node's features can propagate across the graph. In a road network, this corresponds to how much influence surrounding streets (connected streets) have on a given street's highway type. However, higher polynomial degrees do not consistently improve accuracy, although there is trend seen in $K = 1$ to $K = 7$. The fluctuating performance could be due to the model focusing too much on distant nodes, which might be less relevant for classifying a street's highway type. The observed instability in accuracy may relate to the dual characteristics of road networks, which can display both heterophily and homophily. Heterophilic networks consist of connected nodes that possess dissimilar features. In the context of road networks, this would refer to streets that are directly linked but differ in highway classification (for instance, a residential street connecting to a national/primary road) or exhibit contrasting properties. Conversely, homophilic networks consist of interconnected nodes that share similar attributes. For example, streets that are connected may all fall under the same highway type (such as being residential streets) or may have similar characteristics like speed limits or lane counts.

To conclude this section, the experiments performed to analyze the effects of node centrality are composed of four parts: (1) test accuracy of six ML models on three feature sets, (2) the visualization of the node embeddings on the best feature set across all models, (3) the classification metrics for the best feature set, and (4) accuracy, running time behavior of the two models, GraphSAGE and ChebNet, when two parameters are modified. The results showed that there is a significant difference when GNN and traditional ML models are subjected to different feature sets, which can introduce noise and redundancy in the embedding space, just like what [15] found out that using all node features for learning on node classification task leads to sub-optimal performance. The visualization of the embeddings has also provided insight on how classes exhibiting similar characteristics affect the ability of the model to learn the clustering by comparing the distribution of points across the embedding space and the ground truth labels.

# 4 Conclusion and Recommendations

This section discusses the results from training six machine learning models on a road network graph for node classification. The analysis shows that GNN-based models, especially GraphSAGE, perform best in clustering road types, leveraging local neighborhood structures to capture subtle distinctions between road types. The results also emphasize how model performance fluctuates with feature set variations. For example, the top-performing model for S1 (D4) saw significant drops in accuracy when using only centrality features (S2) and combined features (S3), achieving accuracies of 0.48 and 0.59, respectively. These findings suggest that centrality measures alone lack the contextual information required for road classification, and combining them with road features may not be sufficient to achieve high accuracy. While the Naive Bayes (NB) model is not GNN-based, its competitive performance compared to the Graph Attention Network (GAT) highlights that traditional models can still yield useful results under certain conditions. In conclusion, GNN-based models demonstrate promising clustering capabilities for road classification, particularly with GraphSAGE's superior separation of residential, tertiary, and primary road types. The neighborhood aggregation mechanism in GraphSAGE provides a more effective capture of local structural variations, helping distinguish road types that have overlapping physical characteristics but differing functional roles. However, the challenges GNN models face in separating tertiary roads from residential roads and secondary highways from primary roads reveal inherent limitations in encoding global topological contexts. These road types often share structural similarities—such as intersection frequency and lane configuration—but differ in their roles within the overall network, which includes factors like connectivity and surrounding infrastructure. Additional exploration could involve integrating external datasets, such as population density, land use, or traffic data, to enrich the functional context for road types. This approach may enhance model accuracy by allowing for the incorporation of environmental and infrastructural characteristics that contribute to road type distinctions, especially in urban areas with complex road networks.

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

# A   Other Experimental Results

Figures A.1 and A.2 shows the node embedding for feature sets S2 and S3, together with the confusion matrix of each models.

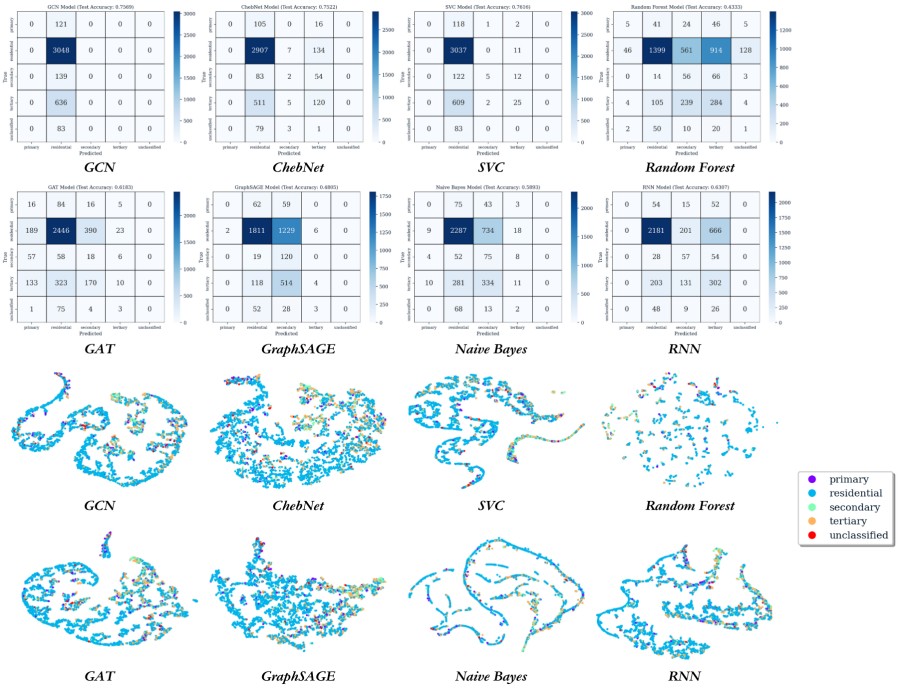

**Figure A.1.** S2 Feature Set: Visualization of the node embeddings learned by the different models.

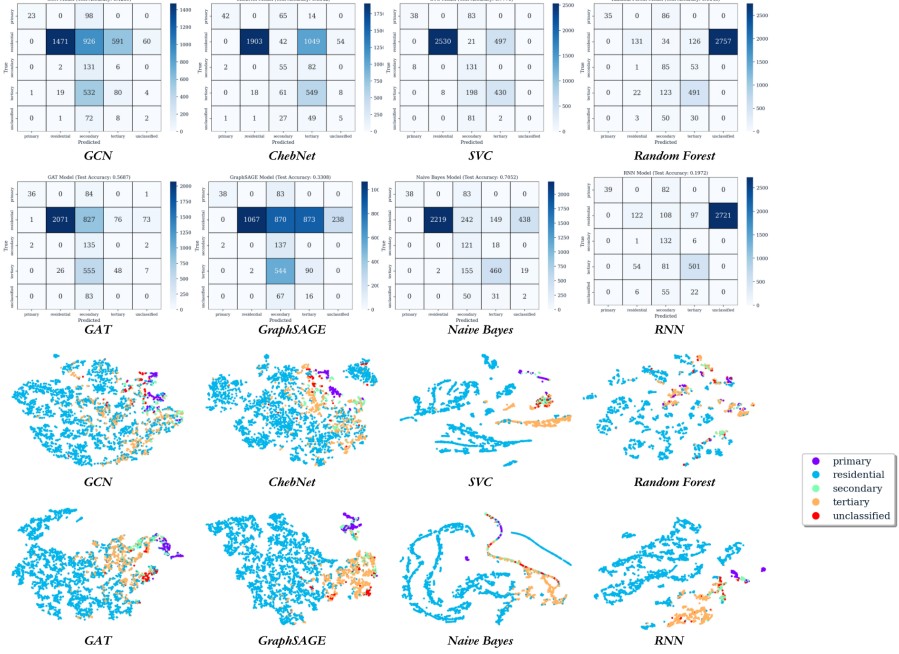

**Figure A.2.** S3 Feature Set: Visualization of the node embeddings learned by the different models.

