# OpenReview forum: "Effects of Node Centrality Measures for Classification Tasks using GNNs"
_NLDL.org/2025/Conference — Submitted to NLDL 2025_

### Official Review · Reviewer_Bavt · 2024-09-24
**Not good enough**

**Confidence:** 2

**Summary:**

This manuscript studies the impact of node centrality measures on the performance of Graph Convolutional Networks (GCNs) in road-type classification. Various node centrality measures, such as betweenness, page rank, closeness, and degree centrality, are investigated. The study reports that using only the primary road features (i.e., road length, width, speed limit, and traffic direction) without additional centrality scores significantly improves classification accuracy.

**Strengths:**

I could not find any strength.

**Weaknesses:**

There are many weak points in this manucsript. I enumerate some of them below.

1. Grammatical Errors and Poor Writing Quality:
The manuscript contains many grammatical errors and poorly written explanations, which makes it difficult to follow. This lack of clarity seriously undermines the readability.

2. Unclear Methodological Explanations:
The methodology section is inadequately described, and key details necessary for the reader to understand the study’s approach are either missing or unclear. Especially, the transformation from primal graph to dual graph is not adequately explained.

3. Lack of Explanation about Difference from Previous Work:
Since the use of GCNs for road type classification is not new, the authors need to explain the difference between their work and existing studies. But a clear explanation is not given.

4. Unclear Results and Conclusions:
The result is inadequately presented and difficult to interpret. Several problems can be raised. (1) The explanation of how the results were obtained is unclear. (2) The discussion around the findings is vague. For example, it is mentioned that excluding node centrality scores improves accuracy, but the rationale behind this is not well articulated. (3) The experimental results supporting the main argument, using only the primary road features gives the best classification accuracy, are weak.

**Justification:**

As discussed in the Weakness section, this manuscript has a lot of flaws such as poor writing quality, unclear methodology explanation, lack of novelty, and weak experimental results. I am certainly not familiar with the topic treated in the manuscript, but I strongly think this kind of low quality stuff should not be accepted.

---

> ### Author Rebuttal · Authors · 2024-10-25
>
> Baseline models and other GNN models are now incorporated in the revised version. Writing is now improved.

---

### Official Review · Reviewer_5H9U · 2024-10-08
**I consider the paper not suited for publication in its current form.**

**Confidence:** 4

**Summary:**

The paper explores graph convolution networks for the prediction of road type based on nodal features encompassing various basic properties of the road such as length and speed limit as well as graph structures in terms of shared intersections with other road segments and enriched with nodal information such as degree, number of shortest paths traversing node etc. The paper finds  that basic features are most important for successful prediction of road type in terms of residential, secondary, tertiary and service type. PCA is further employed on the basic properties and graph derived features finding that the basic properties better accounts for information regarding road type in the PCA subspace.

**Strengths:**

The idea of using GCN and the dual graph exploring road segments as nodes and their intersections as edges is sound.

The approach enriching the nodal features with graph derived properties is interesting.

The prediction of road type seems underexplored in the literature as opposed to other tasks such as traffic prediction.

**Weaknesses:**

When predicting road type I would have expected satellite image data to be highly useful based on GPS coordinates of the roads.This has previously been considered in the context of road quality prediction, see also:
Brewer, Ethan, et al. "Predicting road quality using high resolution satellite imagery: A transfer learning approach." Plos one 16.7 (2021): e0253370.
Would such information be available that could form highly relevant node information not accounted for by the modeling approach. In fact, I would expect high resolution satellite image data to well predict the road type as opposed to relying on basic road features and graph properties. This should be clarified.

Whereas road type prediction has been less commonly explored in the literature GCNs have been widely used for traffic prediction. It is unclear why the authors do not use as starting point architectures designed for traffic prediction and compare the performance of these modeling procedures to their present approach. See also:
Guo, Kan, et al. "Optimized graph convolution recurrent neural network for traffic prediction." IEEE Transactions on Intelligent Transportation Systems 22.2 (2020): 1138-1149.
It would also benefit to clarify what the merits of the present design choices are as opposed to the existing GCNs used in the context of traffic prediction (as opposed to current context of road type prediction), and clarify whether the dual graph representation presently used has previously been explored.

The results are not very convincing and the overall accuracy given on page three does not appear better than chance given the imbalance of the classes. This needs to be discussed. Furthermore, the confusion matrices given in the appendix does not seem to predict some of the classes at all and also exhibit low accuracy (i.e., high degrees of misclassification). Furthermore, no error bars are reported and it is thus unclear how prone the results are to fluctuations wrt. parameter initialization, data splits etc. The significance of the results and the strength of the present approach is thus unclear.

The paper would also benefit from being restructured such that the methods section details the considered model architectures and their motivations. Furthermore, the paper could improve its presentation as also outlined in the minor comments below.

Minor comment

The introduction should be broken into sections and reads currently as one very long paragraph – this will help the reader.

OSM is not explained as abbreviation – do you mean open street map?

The sentence “The specific application of GCN in he context of road network modeling in the Philippines is not evident at the time of writing” – if the application is not evident it is unclear why this study is of interest and importance. Did the authors mean to convey something else with this sentence? – that GCN has not been applied previously in this context?

NE is not explained as abbreviation in “a few authors have applied NE…” – do you mean network embeddings?

“The raw obtained “ -> “the raw data obtained”

QGIS is not explained I assume this refers to the type of geographic information system used.

**Justification:**

The results are not very convincing and the approach should be better motivated. The paper needs to be improved also in its presentation and results statistically assessed at least in terms of error bars. It is also unclear what the benefits are of the proposed approach given that basic features appears to be best - i.e. how would a simple logistic regression model using the basic features or a standard feed forward neural network using the basic features perform? - and is the approach taken in this context meritable?
In summary, I consider the paper not ready for publication in its current form.

---

> ### Author Rebuttal · Authors · 2024-10-25
>
> Baseline models and other GNN models are now incorporated in the revised version. Writing is now improved.

---

### Official Review · Reviewer_fUwb · 2024-10-10

**Confidence:** 5

**Summary:**

In this submission, the authors train graph convolution neural networks to achieve node-level road network classification.
A sophisticated feature extraction method is applied to convert road network data to directed dual multigraph.
A GCN is trained based on the extracted graph, and the authors quantitatively analyze the impacts of different node features (the node importance and the primary node features) on the model performance.

**Strengths:**

1. The problem is interesting, and the dataset is unique to my knowledge.

2. Applying graph neural networks to solve this classification task is reasonable. Converting road networks to dual multi-graphs makes sense to me.

**Weaknesses:**

1. The writing and the organization of this paper are unsatisfying. The title does not specify the road network task, which does not match well with the main content of the paper. The implementation details and the experimental results are not shown in the main pages, and the motivation fo the clustering experiments and the information they released are not explained well. Overall, although the paper is short, its organization and writing are confusing.

2. The experimental part is not solid enough. Basically, the GCNs used in the experiments have the same architecture except for some pre-defined offsets. Why not apply more advanced GNNs, like ChebyNet? In addition, I have no idea why the authors add the offsets to GCNs, especially for the model 3 --- adding any offsets before softmax is meaningless because softmax is shifting invariant. Without any competitive baselines, the rationality of this work is not convincing.

**Justification:**

As I mentioned above, this submission has many holes in its writing, organization, and experiments. Its technical quality and novelty are not high enough.

---

> ### Author Rebuttal · Authors · 2024-10-25
>
> Baseline models and other GNN models are now incorporated in the revised version. Added ChebNet to test the model performance on the node classification task at hand. Writing is now improved.

---

### Official Review · Reviewer_4i7E · 2024-10-12

**Confidence:** 4

**Summary:**

This paper analyzes a road network using GCNs. Specifically, the paper uses a road network datasets from Philippines.

The method the paper uses is as follows. It transforms the network into its dual and then adds new features. This feature graph is inputted into a GCN to obtain labels for the type of the road.

**Strengths:**

The paper works with less explored dataset and application using  graph neural networks.

**Weaknesses:**

I think the paper has a few weaknesses.

1. Novelty. Except for applying an existing technique to an existing problem. I’m not sure what is new. The paper proposes adding new features based on the network properties but then after computing the PCA embedding, decides not to use their new features as you get better clustering without them.

2. Lack of baselines and discussions. The paper then trains the GCN and obtains an accuracy of around 60%. However I do not know how to interpret this without other methods to compare against.

3. The writing could also be improved. There are many acronyms that’s are not defined. Additionally 12 figures for 5 pages is quite a few. I do not think all of them are needed.

**Justification:**

The novelty and significance of the paper is quite limited. It is not clear what new insights a reader is supposed to get.

---

> ### Author Rebuttal · Authors · 2024-10-25
>
> Baseline models and other GNN models are now incorporated in the revised version. Writing is now improved.

---

### Meta-Review · Area_Chair_V6qX · 2024-10-28

**Recommendation:** Reject
**Confidence:** 5

**Metareview:**

The initial version of this paper proposed to use GCN to classify road types using a particular dataset from Philippines. As pointed out by the reviewers, the paper suffers from a number of issues such as
- questionable novelty: while the particular task studied in the paper (road type classification) is not commonly found in the litterature, there are a number of relevant papers/techniques pointed out by the reviewers that have not been taken into account.
- results lacking comparison to a baseline and showing relatively low performance.
- methodology described too vaguely and lacking details
- poor structure of the paper

The reviewers were quite unanimous about the lack of novelty and insufficient quality of the initial paper, and did not recommend its acceptation. In addition to that, the authors did not make the effort of replying to the points raised by the reviewers during the rebuttal.

The authors however submitted a revised version of the paper that was *entirely* rewritten (a comparison between the two versions reveals the lack of intersection). This raises a number of issues, as the reviews submitted become less relevant and makes it difficult to see whether the observations were taken into account.

Concerning the revised version of the paper, after getting feedback from the reviewers:
- The overall structure of the paper has been improved.
- Some comparison with existing relevant litterature for the considered task is still missing. Results are missing a refernence to a baseline from the litterature applied to the same dataset.
- The authors spend a lot of time detailing classic GNN constructions, and explaining ChebNet details (which are well-known). The fact that they use the dual graph representation should have been more detailed as this is a crucial point.
- Eq. 3 show they are introducing self-loops (albeit without justification), and IMO is not correct  as D should be the degree matrix of A+I.
- The new results presented are strange: the authors train several models using road features (S1), centrality measure only (S2) or both road features and centrality (S3). While it could be understandable to see a performance difference between S1 and S2, the performance of S3 should at least be as good as the one from S1. However table 2 shows that there are multiple cases (D2, D3, D4, B3) where the performance of S3 is lower than S1. This suggests that there might be experimental issues. The analysis of the results did not provide any explanation regarding this (while IMO this raises string concerns about the validity of experiments)
- Details about the dataset structure (class repartition) that were present in the initial submission have been removed. Since this is not a well-known dataset, this should have been kept in order for the reader to better understand the results
- Confusion matrices in appendix are missing results about the S1 version, and the comments by reviewer 5H9U remain valid "the confusion matrices given in the appendix does not seem to predict some of the classes at all and also exhibit low accuracy (i.e., high degrees of misclassification)"

In conclusion, while the paper has improved, there are still too many open issues with the experiments and results to accept it, I recommend its rejection.

**Suggested Changes To The Recommendation:**

2: I'm certain of the recommendation.  It should not be changed

---

### Decision · Program_Chairs · 2024-11-06

Reject